# Underwater Acoustic Signal Detection Using Calibrated Hidden Markov Model with Multiple Measurements

**DOI:** 10.3390/s22145088

**Published:** 2022-07-06

**Authors:** Heewon You, Sung-Hoon Byun, Youngmin Choo

**Affiliations:** 1Department of Ocean Systems Engineering, Sejong University, Seoul 05006, Korea; youhw0905@sju.ac.kr; 2Korea Research Institute of Ships and Ocean Engineering (KRISO), Daejeon 34103, Korea; byunsh@kriso.re.kr; 3KRISO Campus, Korea University of Science and Technology, Daejeon 34103, Korea; 4Department of Defense Systems Engineering, Sejong University, Seoul 05006, Korea

**Keywords:** sonar signal detection, hidden Markov model, genetic algorithm

## Abstract

It is important to find signals of interest (SOIs) when operating sonar systems. A threshold-based method is generally used for SOI detection. However, it induces a high false alarm rate at a low signal-to-noise ratio. On the other side, machine-learning-based detection is performed to obtain more reliable detection results using abundant training data, costing intensive time and labor. We propose a method with favorable detection performance by using a hidden Markov model (HMM) for sequential acoustic data, which requires no separate training data. Since the detection results from HMM are significantly affected by the random initial parameters of HMM, the genetic algorithm (GA) is adopted to reduce the sensitivity of the initial parameters. The tuned initial parameters from GA are used as a start point for the subsequent Baum–Welch algorithm updating the HMM parameters. Furthermore, multiple measurements from arrays are exploited both in determining the proper initial parameters with GA and updating the parameters with the Baum–Welch algorithm. In contrast to the standard random selection of the initial point with single measurement, a stable initial point setting by the GA ensures improved SOI detections with the Baum–Welch algorithm using the multiple measurements, which are demonstrated in passive and active acoustic data. Particularly, the proposed method shows the most confidential detection in finding weak elastic surface waves from target, compared to existing methods such as conventional HMM.

## 1. Introduction

Sonar systems with arrays comprising multiple sensors have been used to detect signals of interest (SOIs). However, various noises from vessels, submarines, and fish schools, etc., exist in the ocean and are measured by the sensors along with the SOIs. Therefore, detection methods are required to discriminate the desired signals from the noises.

Threshold-based detection schemes such as energy detection [1] or constant false alarm rate (CFAR) detection [2] are generally used in finding the SOIs. In energy detection, the energy of measured data is compared with a predefined threshold value. CFAR detection is a scheme that uses an adaptive threshold based on the relationship between a specified cell (sample under test) and adjacent auxiliary data. The threshold-based detections do not require prior information regarding the marine environment and exhibit low computational complexity. However, the detection performance is inferior in low signal-to-noise ratio (SNR) owing to the simple decision rules for SOIs. Hence, sophisticated detection methods using algorithms from machine learning (ML) have been proposed [3,4,5,6].

Owing to technological developments, various ML schemes have been applied in detecting the SOIs passively or actively [3,4,5,6,7], which treat the SOI detection as classification problems. To distinguish a target from a clutter in active sonar systems, a perceptual-based signal features from the human auditory system are exploited [3]. To suppress interference from background noise in recognizing underwater sound signals, a denoising autoencoder are used with random forest [4]. Furthermore, signal detection methods using various convolution neural networks have been used actively [5,6,7]. Although ML-based detections or classifications have remarkably enhanced performance, they require abundant training data for a given task, which have time and labor costs.

A hidden Markov model (HMM), a ML algorithm, has been applied widely to speech and text recognition with sequence data [8,9,10,11]; it estimates hidden states (or hidden information) of samples in the sequence data by using probabilities (HMM parameters) explaining the hidden states, which are extracted from the given data themselves. HMM has been applied to sequential data measured by radar and sonar systems to detect the corresponding target signals (i.e., SOIs) [12,13,14]. For a track-before-detection strategy in noisy environments, HMM was used for radar target detection to avoid the usage of threshold-based detection [12]. In bioacoustics, the vocalizations of Bryde’s whales were automatically identified by using HMM, which enables the SOI detection even when ship noise interfering with the whale sounds is present [13]. In [14], to enhance detection performance in passive sonar data, a pregrouping of acoustic signals (i.e., samples are preliminarily clustered into “signal” and “noise”) was incorporated with HMM.

In the current study, we attempt to detect SOIs in sonar data such as scattered signals from targets with low false alarms using limited measurements. Thus, we propose an HMM-based detection requiring no separate training data. Since the detection results from HMM are significantly affected by random initial parameters of HMM, a genetic algorithm (GA) is utilized to reduce the sensitivity of the initial parameters to the detection [15]. Furthermore, multiple measurements from array are exploited in the HMM-based detection to enhance accuracy and stability in finding the SOIs within sonar data. Section 2 presents problems in underwater signal detection and describes the parameters in the HMM. In Section 3, the proposed scheme using the HMM is explained comprehensively. The detection performance of the proposed scheme is investigated using synthetic passive and real active sonar data (Section 4 and Section 5). Finally, the conclusions are provided in Section 6.

## 2. Problem Description

Here, an HMM-based detection scheme is proposed to detect SOIs with less false alarms and without separate training data by exploiting sequential acoustic data with pre-established probability models in the HMM. The conventional HMM is modified to accommodate multiple measurements from an array, which enhance the detection accuracy and robustness by correlated SOIs over sensors in the array.

The HMM has been widely adopted in speech and text recognition involving sequential data [8,9,10,11]. In the HMM framework, samples in the sequential data have hidden states, which are estimated from observed signals. Sonar data can be sequenced based on the regularity of the SOIs, and the HMM can be applied to underwater acoustic signal detection using time-domain sonar data after quantization is performed. Two states exist in sonar data, i.e., signal and noise states.

Figure 1 shows the structure of the HMM. The HMM (or probability models in the HMM) can be expressed as θ=[π,A,B], where π, A, and B are the initial state distribution, transition matrix, and emission matrix, respectively. The initial state distribution, π, indicates the probability distribution over the states at the initial time. The states, as time progresses, are connected by the first-order Markov chain [16], and their transitions are represented by the transition matrix A (relevant to dotted line), whose element (i,j) represents the probability of the state changing from the *i*th state at the present time to the *j*th state at the next time. The probability of a specific observation at a certain state (emission probability) is represented by emission matrix B (relevant to dashed line). The sizes of the transition and emission matrix are M×M and M×N, respectively, when the number of states is *M* and the values in sonar data are quantized with *N* (in the current study, M=2 and N=150).

In the HMM framework, optimal probability models (also, referred to as optimal parameters) are obtained with best explaining the observations by the Baum–Welch algorithm in HMM [17] ($\theta^{*}=\underset{\theta }{\mathrm{argmax}}P\left(o|\theta \right)$). Then, the hidden states are revealed by the Viterbi algorithm, which uses the estimated parameters with observations as follows [18,19]: $\underset{q}{\mathrm{argmax}}P\left(q|o,\theta^{*}\right)$. From the perspective of detection, the SOIs are identified by samples possessing the “signal” state.

During parameter estimation using the HMM, the Baum–Welch algorithm was used to determine $\theta^{*}$ from randomly selecting the initial values. However, the estimates strongly depend on the initial values owing to local optimal points; HMM-based detection results for the same data can be different owing to the random initial values differing along with the applications of HMM. Hence, several studies have been conducted to determine the appropriate initial values for the parameters [14,15,20,21]. In relevant studies pertaining to the detection of underwater SOIs [14], k-means and pre-grouping were used to derive the proper initial values instead of the random ones. This detection scheme is referred to as expectation-maximization (EM)-Viterbi Algorithm (VA) as in [14]. Although the EM-VA provides an accurate detection of SOIs in environments containing transient noise, its performance degrades when ambient noise is present, as discussed in Section 4. Furthermore, when multiple SOIs exist, SOIs having magnitudes similar to or less than the magnitude of noise are overlooked by the EM-VA.

In the EM-VA [14], a single measurement (acoustic data at a single sensor) was used as observation. In the sonar system, multiple measurements can be acquired by sensor arrays. Unlike previous studies using HMM [14,22,23], here, multiple measurements were exploited not only to determine the reliable initial values using the genetic algorithm (GA) but also to update parameters using the Baum–Welch algorithm; these are described comprehensively in the following section.

## 3. HMM Calibration and Parameter Adjustment Using Multiple Measurements

The Baum–Welch algorithm is sensitive to the initial values of the parameters, and its estimations strongly depend on the initial values. To determine the appropriate initial values, multiple measurements from arrays in the sonar system are used to ensure accurate and robust SOI detection. The detection process, which involves various algorithms, is illustrated in Figure 2. First, the initial values for the HMM, which exhibit the most effectively the observations of multiple measurements in terms of probability, are determined using the GA (Section 3.1). Next, the parameters in the calibrated HMM are adjusted by the Baum–Welch algorithm using multiple measurements (Section 3.2). Finally, the hidden states are derived for the multiple measurements using the Viterbi algorithm, and they indicate the SOIs.

### 3.1. Initialization: Calibrating HMM

We used a GA, which is inspired by the natural selection process, to estimate the initial values of the parameters. The GA is a representative scheme for solving optimization problems, where genes (candidates for the solution) in a population at the current generation produce genes in a population at the next generation using crossover, mutation, and selection (evolution) to approach the solution [24]. In particular, mutation in the GA prevents a solution from being a local optimum.

In the current study, genes in the GA are parameters with distinct values ($\theta_{g}^{p}$, where the subscript *g* and superscript *p* indicate the gene and generation numbers, respectively), and the appropriate initial values for the measurements are determined by evolving $\theta_{g}^{p}$ in the GA. The criterion for the appropriateness is calculated using the fitness function $P_{g}^{p}=lnP\left(o|\theta_{g}^{p}\right)$, where $o=[o_{1},...,o_{T}]$ is observed in the time domain, $o_{t}$ is a quantized acoustic signal $v_{n}$, and *T* is the total number of observations with a signal length. $P(o|\theta_{g}^{p})$ is the likelihood function, which stochastically explains the observation based on the specified parameters and is calculated using the forward or backward algorithm in the Baum–Welch algorithm. In this study, the forward algorithm was adopted for the calculation with probability $P(o_{1},...,o_{t},q_{t}=s_{m}|\theta )$, which was obtained from $\alpha_{t}\left(m^{\prime }\right)=b_{m^{\prime }}\left(o_{t}=v_{n}\right)\sum_{m=1}^{M}\alpha_{t-1}\left(m\right)a_{m,m^{\prime }}$. $q_{t}$ is the observed state at time *t* and is one of the state types $s_{m}$ (i.e., “signal” or “noise”; hence, M=2); $a_{m,m^{\prime }}$ ($\left(m,m^{\prime }\right)$ element of the transition matrix) and $b_{m^{\prime }}$$\left(o_{t}=v_{n}\right)$ ($\left(m^{\prime },n\right)$ element of the emission matrix) are the probabilities of state transition from state *m* to state $m^{\prime }$ and the observation of $o_{t}=v_{n}$ at state $m^{\prime }$, respectively.

A new gene was created for the next population by selectively using $\theta_{g}^{p}$ among the current population; $\theta_{g}^{p}$ with a high probability was used preferentially. This process was repeated until a terminal condition was satisfied.

The signals received at the sensor array contain the SOIs. Hence, the parameters for the measurements are expected to be the same. The shared parameters were estimated to increase the detection accuracy and robustness, and the fitness function for the single measurement was modified to accommodate the multiple measurements with the assumption of their stochastic independence. A product of likelihood functions for the measurements was used for the fitness function [15], as follows: (1)$$P_{g}^{p}=\sum_{k=1}^{K}lnP\left(o^{\left(k\right)}|\theta_{g}^{p}\right)$$

The superscript *k* in parenthesis represents the measurement number; *K* represents the total number of the measurements, which is the same as the number of sensors in the arrays. The optimal values from the GA using (1) are the initial values for the parameter in the HMM, which is referred to as the calibrated HMM herein.

### 3.2. Parameter Adjustment Using Baum–Welch Algorithm with Multiple Measurements

Optimal parameters ($\theta^{*}$) were derived using the Baum–Welch algorithm [17], which uses the parameters from GA ($\theta^{0}$) as a starting point. The Baum–Welch algorithm has an iterative loop composed of an expectation (E) step and a maximization (M) step. Hidden variables from the E-step are used to update old parameters in the M-step, and they are denoted as follows [17]:(2)$$\begin{array}{c} \gamma_{t}=P\left(q_{t}=s_{m}|o,\theta^{q}\right), \end{array}$$
(3)$$\begin{array}{c} \xi_{t}\left(m,m^{\prime }\right)=P\left(q_{t}=s_{m},q_{t+1}=s_{m^{\prime }}|o,\theta^{q}\right), \end{array}$$
where $\theta^{q}$ is the parameter after *q* iterations; the superscript indicates the iteration number. $\gamma_{t}$ and $\xi_{t}$ are the probabilities of state $s_{m}$ at time *t*, and the joint states of $s_{m}$ at time *t* and $s_{m^{\prime }}$ at time t+1 for o and $\theta^{q}$, respectively.

When applying the Baum–Welch algorithm with a single measurement, the parameters are updated using the hidden variables as follows [17]: (4)$$\begin{array}{c} \pi_{m}^{q+1}=\gamma_{1}\left(m\right),\;\;\;\;\;1\leq m\leq M \end{array}$$
(5)$$\begin{array}{c} a_{m,m^{\prime }}^{q+1}=\frac{\sum_{t=1}^{T-1}\xi_{t}\left(m,m^{\prime }\right)}{\sum_{t=1}^{T-1}\gamma_{t}\left(m\right)},\;\;\;1\leq m\leq M,1\leq m^{\prime }\leq M, \end{array}$$
(6)$$\begin{array}{c} b_{m}^{q+1}\left(v_{n}\right)=\frac{\sum_{t=1}^{T}I_{o_{t}=v_{n}}\gamma_{t}\left(m\right)}{\sum_{t=1}^{T}\gamma_{t}\left(m\right)},\;1\leq m\leq M,1\leq n\leq N, \end{array}$$
where $\pi_{m}^{q+1},a_{m,m^{\prime }}^{q+1}$, and $b_{m}^{q+1}\left(v_{n}\right)$ are elements of the initial state distribution, transition matrix, and emission matrix at q+1 iterations, respectively. $I_{o_{t}=v_{n}}$ is an indicator function, which equals one when $o_{t}=v_{n}$. Otherwise, it is zero. *T* is the signal length of observation (or measurement) o. The initial state distribution of (4) is obtained from $\gamma_{1}$. The transition probability of (5) is a conditional probability that accounts for the state changing from the *m*th state at the present time to the $m^{\prime }$th state at the next time. It is the ratio of the sum of joint probabilities of $s_{m}$ at time *t* and $s_{m^{\prime }}$ at time t+1 to the sum of probabilities of $s_{m}$ at time *t*; the sums are conducted in the time domain. The emission probability of (6) conforms to its definition (i.e., the probability of a specific observation quantity $v_{n}$ at state $s_{m}$) by counting $\gamma_{t}\left(m\right)$ with the observation $o_{t}$ matching $v_{n}$ among all $\gamma_{t}\left(m\right)$. Equations (2)–(6) are used repeatedly until the parameters converge or the iteration reaches a predefined number.

To exploit the commonality (i.e., the shared parameters) of the multiple measurements from the array, the parameters are updated during the iterations of the Baum–Welch algorithm as follows [17]: (7)$$\begin{array}{c} \pi_{m}^{q+1}=\frac{1}{K}\sum_{k=1}^{K}\gamma_{1}^{\left(k\right)}\left(m\right),\;\;\;\;\;1\leq m\leq M \end{array}$$
(8)$$\begin{array}{c} a_{m,m^{\prime }}^{q+1}=\frac{\sum_{k=1}^{K}\sum_{t=1}^{T^{\left(k\right)}-1}\xi_{t}^{\left(k\right)}\left(m,m^{\prime }\right)}{\sum_{k=1}^{K}\sum_{t=1}^{T^{\left(k\right)}-1}\gamma_{t}^{\left(k\right)}\left(m\right)},\;\;\;1\leq m\leq M,1\leq m^{\prime }\leq M, \end{array}$$
(9)$$\begin{array}{c} b_{m}^{q+1}\left(v_{n}\right)=\frac{\sum_{k=1}^{K}\sum_{t=1}^{T^{\left(k\right)}}I_{o_{t}^{\left(k\right)}=v_{n}}\gamma_{t}^{\left(k\right)}\left(m\right)}{\sum_{k=1}^{K}\sum_{t=1}^{T^{\left(k\right)}}\gamma_{t}^{\left(k\right)}\left(m\right)},\;1\leq m\leq M,1\leq n\leq N, \end{array}$$

Hidden variables $\gamma_{t}^{\left(k\right)}$ and $\xi_{t}^{\left(k\right)}$ are calculated using the *k*th measurement $o^{\left(k\right)}$ in the E-step; $T^{\left(k\right)}$ is the signal length of $o^{\left(k\right)}$ and is set as a constant of *T* in the current study. Equations (7)–(9) are obtained by modifying (4)–(6) with an additional summation over the spatial domain based on multiple measurements by the array. The initial state distribution of (7) is the average of $\gamma_{1}^{\left(k\right)}$ over the spatial domain. Similar to (5) and (6), the transition probability of (8) is the ratio of the sum of the joint probabilities $\xi_{t}^{\left(k\right)}\left(m,m^{\prime }\right)$ and the sum of the corresponding marginal probabilities $\gamma_{t}^{\left(k\right)}\left(m\right)$; the sums are conducted in the space and time domains. The emission probability of (9) is calculated by counting $\gamma_{t}^{\left(k\right)}\left(m\right)$, with the observation $o_{t}$ matching $v_{n}$ among $\gamma_{t}^{\left(k\right)}\left(m\right)$.

Multiple measurements from the array are beneficial to the HMM because they provide additional samples that are in proportion to the number of sensors for estimating the conditional probabilities, as shown in (7)–(9). The probabilities from rich data are more reliable and result in stable and accurate signal detections.

Next, the hidden states for each measurement are revealed using the Viterbi algorithm [18,19], based on observations as well as shared parameters $\theta^{*}$ from the Baum–Welch algorithm. ${\hat{s}}^{\left(k\right)}$, which comprises hidden states as time progresses at the *k*th measurement, is derived using the Viterbi algorithm by maximizing $P(q^{\left(k\right)}=s^{\left(k\right)}|o^{\left(k\right)},\theta^{*})$; it indicates the SOIs in the measurement. The suggested process is abbreviated as the GA-HMM.

Although the parameters can be determined using the GA or Baum–Welch algorithm separately, the two optimization schemes are used sequentially in SOI detection for a superior estimation of parameters; the GA derives an unbiased initial point for the Baum–Welch algorithm (Figure 3a), and a desired optimal point is subsequently determined from the initial point (Figure 3b). The detection performance afforded by the Baum–Welch algorithm alone is sensitive to the random initial values of the parameter (or random initial point), which are updated consecutively using the Hill-Climbing [25,26] and hence can fall into a local optimum point next the neighboring random initial point. Using only the GA incurs a high computational cost for parameter convergence. Furthermore, noise hinders the GA from converging near global optimal points. The parameters cannot converge even after sufficient generations; hence, the detection performance based on the GA deteriorates.

## 4. Analysis of GA-HMM Using Synthetic Data

The detection performance of the GA-HMM was analyzed by comparing its detection results with those of other schemes. The effects of the fine initial point from the GA were demonstrated with synthetic data.

### 4.1. Numerical Environment

To analyze the GA-HMM, synthetic data were generated while considering the acoustic signals measured using the sonar systems. Each synthetic datum with a signal length of 0.3 s was discretized with a sampling frequency of 500 Hz and contained 150 samples (T=150). Here, the SOI in the synthetic data was a 50 Hz three-cycle sine wave comprising 30 samples, and it was contaminated by additive white Gaussian noise (Figure 4a). Although the starting point of the SOI did not affect the detection performance, the 50th sample of the synthetic data was used as the starting point to ease the visual inspection of the detection results.

Noise with various magnitudes were added to the clean synthetic to investigate the detection performance according to SNRs. Additionally, different numbers of the synthetic data were used for the detection to demonstrate properties of multiple measurements in finding the SOI.

In the current study, an observation value of HMM is an absolute value of the acoustic signal quantized with uniform intervals of 150 (N=150) after normalization with its absolute maximum. To obtain the fine initial point using the GA, 200 randomly generated parameters were used as genes in the first-generation population. A score for the appropriateness was calculated for the genes using the fitness function presented in (1), and the genes with high scores had a high probability of being selected for generating the next genes with crossover. The probability of mutation was set to 0.01. The most feasible parameter after 10 generations (p=10) was used for the fine initial point. In the GA, the transition probability between the same (or different) states had a lower (or upper) bound of 0.5, owing to rare transitions between different states, which occurred at the 50th (from noise to signal) and 80th samples (from signal to noise) among 150 samples in the simulation.

Subsequently, the Baum–Welch algorithm commenced from the fine initial points and terminated when the parameters converged or the iteration reached a predefined number (in this study, Q=500).

The Viterbi algorithm, which is applied to the sequential samples in the observations with estimated parameters, implies the states at the samples with a value of 0 or 1 (M=2). The variance of samples with the same state was calculated. The state with a higher (or lower) variance was assigned to the “signal” state (or “noise” state) as in [14]; here, samples with values of 0 and 1 correspond to the “noise” and “signal” states, respectively. Quantities for the hyperparameters in the GA-HMM, including the quantization number and upper and lower bounds, were determined empirically.

### 4.2. Detection Performance Analysis of GA-HMM

Figure 4 shows representative examples of detection results obtained by the GA-HMM, EM-VA, and Random-HMM; the HMM using a random initial point for the Baum–Welch algorithm is referred to as the Random-HMM here in. The SOI is indicated by vertical dashed lines. The SOI was detected under a harsh condition without using a matched filter enhancing the SNR (passive sonar signal detection). Noise comparable to the SOI (SNR = 8 dB) restricts the use of threshold-based detection schemes. Therefore, sophisticated schemes were used. In this study, the GA-HMM and Random-HMM were applied to perform detections using single or multiple measurements based on (1) and (7)-(9). Meanwhile, the EM-VA used a single measurement to identify the SOI because it cannot accommodate multiple measurements [14].

When using the EM-VA, most of the samples in the synthetic data were identified as the SOI by noise, and false alarms occurred, as shown in Figure 4b. Figure 4c,d show the detection results obtained from the Random-HMM and GA-HMM based on a single measurement. Many noise samples were misclassified as SOIs, increasing false alarm rates (FAR) owing to some inappropriate initial values in Random-HMM. This problem was mitigated using the GA-HMM, which determined the parameters using the fine initial point. However, considerable false alarms remained. Therefore, multiple measurements comprising 30 synthetic data were used, as shown in Figure 4e,f, to reduce false alarms. As a result, the Random-HMM using multiple measurements achieved significantly reduced the FAR of the SOI sample. On the other hand, the GA-HMM using multiple measurements exhibited the highest recall with less false alarms, thereby demonstrating its superior detection performance compared with the considered schemes.

Table 1 summarizes the recall, FAR, and computation time of the schemes based on the average detection results for 100 trials at a fixed SNR of 8dB. In this study, recall is defined based on the ratio of the number of correctly identified SOI samples to the total number of SOI samples, and the FAR is defined based on the ratio of the number of misidentified noise samples to the total number of noise samples. Although the EM-VA exhibited a high recall, it incorrectly identified noise persistently. In particular, the noise near the SOI tended to be identified as a “signal”, and it resulted in the highest FAR as shown in Figure 4b. The Random-HMM using single measurement overlooks the SOI and misclassified noise because of unstable detection from the random initial point; thus, it resulted in an inferior recall and FAR. While these problems were alleviated using the single-measurement GA-HMM, it still exhibited a considerable FAR. Therefore, all single-measurement schemes exhibited unsatisfactory detection performance owing to excessive false alarms. Noise could not be distinguished from the SOI, thereby resulting in high FARs in the scarce measurement. Therefore, multiple measurements were used to mitigate these problems.

The considered schemes were implemented at a computer with an intel(R) Core (TM) i9-9900K CPU, and the corresponding computational times were measured (Table 1). Although the proposed scheme showed the hugest computational burden, it can be applied to acoustic measurements during experiments and detect SOIs in semi-real-time, owing to its computational time in the order of 10 s.

The multiple-measurement Random-HMM exhibited improved performance in terms of both recall and FAR (moderate recall with significantly reduced FAR) because it exploited the consistency of the SOI in the multiple measurements when updating the parameters and was less affected by erratic noise. Detection performance was significantly improved by using multiple measurements to determine better initial points with GA and update parameters with the Baum–Welch algorithm. As a result, the multiple-measurement GA-HMM exhibited the highest recall and lowest FAR, indicating that most of the samples were identified correctly.

To investigate the detection performance of the schemes for various noise magnitudes, synthetic data with various SNRs were generated. The recalls and FARs from schemes were displayed according to the SNRs, as in Figure 5, where they were averaged over 100 trials at each SNR. At low SNR, EM-VA had high recall and FAR, and most of samples were identified as “signal”. With the increment in the SNR, the false alarm significantly reduced with lower recall by overlooking SOI samples more frequently. The other schemes using single measurement resulted in increased recalls and decreased FARs as the SNR increased. In particular, the initial point obtained using the GA improved the single-measurement performance, which also improved as the SNR increased. The schemes based on single measurement could not provide reliable detections (even at high SNRs) because of their insufficient recalls (EM-VA and Random-HMM) or high FARs (GA-HMM and Random-HMM). As shown previously in Figure 4, the multiple measurements improved the detection performance by evaluating the conditional probabilities in (7)–(9) more confidently. The recall (or FAR) of the multiple-measurement Random-HMM improved gradually as the SNR increased and reached 0.75 (or approximately 0.2) at a high SNR of 13. The detection performance improved considerably by the multiple-measurement GA-HMM, whose classification accuracy was accelerated by the increase in the SNR and became almost perfect at the appropriate SNR. Additional methods for obtaining fine initial points such as the GA are important in HMM-based detection because the initial points significantly affect the parameter estimation in the Baum–Welch algorithm of HMM.

Figure 6 shows the detection results from the Random-HMM and GA-HMM analyzed based on the measurement number (sensor number in array) at a fixed SNR of 8 dB. Although their performances improved in proportion to the measurement number, the GA-HMM with a fine initial point exhibited superior accuracy in terms of detection regardless of the measurement number. The GA-HMM exploited multiple measurements more effectively than the Random-HMM because it used them in the Baum–Welch-algorithm-based update as well as the GA-based initial point identification. The performance differences increased until the measurement number reached 30. Despite the slow performance enhancement after 30 measurements, the multiple measurements afforded accurate and robust SOI detection.

## 5. Application of GA-HMM to Measured Acoustic Data

The feasibility of the multiple-measurement GA-HMM was investigated by analyzing acoustic data from a water tank experiment, which included intense specular echoes and weak elastic waves from shell targets (SOIs). The detection results for the real data were compared with those obtained using the EM-VA and multiple-measurement Random-HMM.

### 5.1. Experimental Environment

An experiment for target scattering was conducted in a water tank with size of 35 m (length) × 20 m (width) × 9 m (depth). A simple illustration of the water tank is shown in Figure 7a, and its details are provided comprehensively in [27]. A 1 s long linear frequency-modulated pulse signal with a bandwidth between 0.5 and 25 kHz from a transducer impinged on the cylindrical shell target, and the scattered signal from the target was measured using two hydrophones at different water depths (referred to as R1 and R2). After applying a matched filter to the measured signals (pulse compression), a specific time period of 1.5 ms, including the returns from the target (intense specular echo and two subsequent weak elastic surface waves) was selected, as illustrated in Figure 7b, and the corresponding observation size *T* was 150, with a sampling frequency of 100 kHz.

The specular echoes from two measurements exhibited similar amplitudes and arrival times (approximately 0.5 ms) and were insensitive to the depth difference. The elastic waves, which exhibited distinct circumferential paths on the cylinder surface, were not consistent with the measurements in terms of amplitudes and time delays. A slight gap existed between the first elastic wave (approximately 0.8 ms) and the specular echo in the R1 measurement. On the other hand, the first elastic wave (approximately 0.6 ms) was immediately behind the specular echo and exhibited a small amplitude in the R2 measurement. The arrival times were confirmed by comparing the measured data with the simulated data based on the same environment [27]. The second elastic waves in the R1 and R2 measurements arrived at approximately 1.3 and 1.5 ms, respectively. The detection performance of the schemes was analyzed in terms of the identification of weak elastic waves, as will be described in the following subsection.

### 5.2. Detection Results of GA-HMM for Measured Acoustic Signals

Figure 8 shows the detection results from the EM-VA, the multiple-measurement Random-HMM, and the multiple-measurement GA-HMM. While the EM-VA was applied to the measurements individually, the Random-HMM and GA-HMM detected the SOIs after the shared parameters over the measurements were estimated. In the EM-VA, k-means and pre-grouping were applied to evaluate the initial values, and obtain a consistent detection result for the same data. However, although the GA-HMM is less affected by the random initial value used in GA, the detection of the GA- and Random-HMM varies depending on the trials, even if the same data are used. Hence, the Random–HMM and GA-HMM were applied to the measurements repeatedly, and the average of 100 detection results were used. The value for the state of a certain sample in the measurements was between 0 and 1, and a sample with a higher (or lower) number was likely to be the signal (or noise). In this study, samples exhibiting values exceeding 0.7 and less than 0.3 were classified as “signal” and “noise”, respectively. The remaining samples exhibiting values between 0.3 and 0.7 were neither “signal” nor “noise” and were referred to as “unclear samples”.

The EM-VA detected the specular echoes without unclear samples owing to the consistent estimation, whereas it overlooked the weak elastic waves in both measurements. Furthermore, the elastic waves having similar magnitudes with noise made the Random-HMM using multiple measurements determine most samples as unclear samples, except for some samples within the specular echoes, and it was detrimental to identifying the SOIs. The GA-HMM using multiple measurements also suffered from detecting the weak elastic waves. Particularly, the second elastic wave in the R2 measurement was misidentified as “noise”. In the water tank experiment, acoustic signals were measured by two hydrophones, and thus sparse measurements were used for the detection, which resulted in the diminished detection performance, compared to those using the synthetic data. The performance reduction could be mitigated by using additional measurements, which were unavailable in the current study. However, the multiple-measurement GA-HMM showed the best performance among the considered schemes. It significantly reduced the unclear samples and detected the specular echoes and elastic waves most confidently; even unclear samples locating between noise provided clues for the SOIs (e.g., the first elastic wave in R2 measurement).

Among the considered schemes, the GA-HMM exhibited the best signal detection and false alarm reduction in both the synthetic and measured data. Hence, the GA-HMM is applicable to sonar signal detection when ML-based schemes are unavailable because of inadequate training data.

## 6. Conclusions

We proposed a novel HMM-based detection method to accurately identify signals with a low FAR without requiring training data. However, the Baum–Welch algorithm for parameter estimation in the HMM is sensitive to the initial point and the problem of falling into the local optimum point often occur because of a random initial point. The GA provided a proper initial point for obtaining a global optimal point and determined the appropriate parameters by using the Baum–Welch algorithm with the initial point.

Furthermore, by using multiple measurements both in deriving the initial point with GA and updating the parameters with Baum–Welch algorithm, SOIs are detected more accurately and reliably; GA and multiple measurements improve the stability and accuracy of SOI detection, respectively. Thus, the multiple-measurement GA-HMM displayed superior performance in passive and active acoustic data, which are from simulation and real measurements, respectively. The detection results are compared with those from other detection schemes such as EM-VA and Random-HMM. Particularly, inconsistent and unclear detections from conventional HMM (single-measurement Random-HMM) are significantly alleviated by the multiple-measurement GA-HMM at the cost of computational complexity.

## Figures and Tables

**Figure 1 sensors-22-05088-f001:**
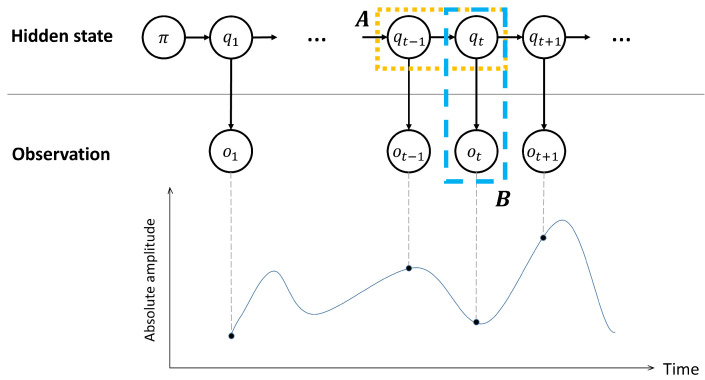
The structure of the HMM. The HMM explains observations $o_{t}$ by using three probability models. An initial state distribution π is the probability distribution over the states at the initial time. The state change (marked with dotted line) is accounted for by the first-order Markov chain, and the previous state affects the present state. An observation at a specific time appears probabilistically, which depends on the state (marked with dashed line). HMM finds the optimal probability models for observations (Baum–Welch algorithm), which are subsequently used to identify the hidden states with the observations (Viterbi algorithm).

**Figure 2 sensors-22-05088-f002:**
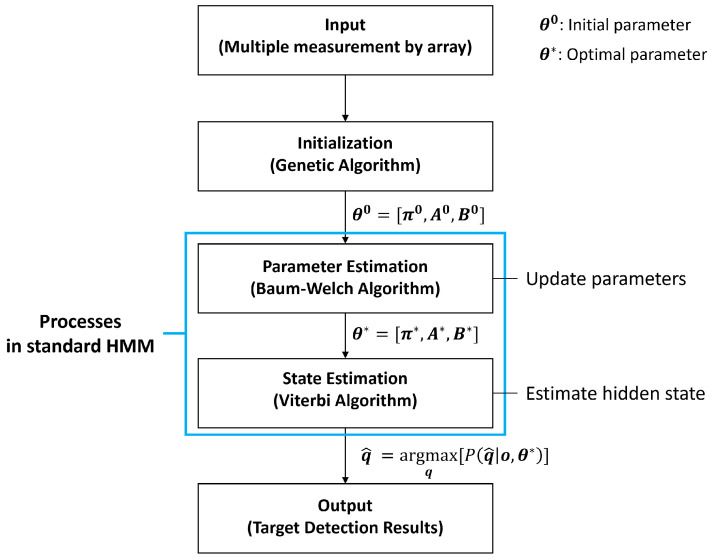
Block diagram of detection process in hidden Markov model (HMM). Multiple measurements were exploited to determine the best initial values for the HMM parameters using the genetic algorithm. They were updated with the Baum–Welch algorithm using the multiple measurements. Then, hidden states of the multiple measurements were revealed via the Viterbi algorithm using the HMM parameters.

**Figure 3 sensors-22-05088-f003:**
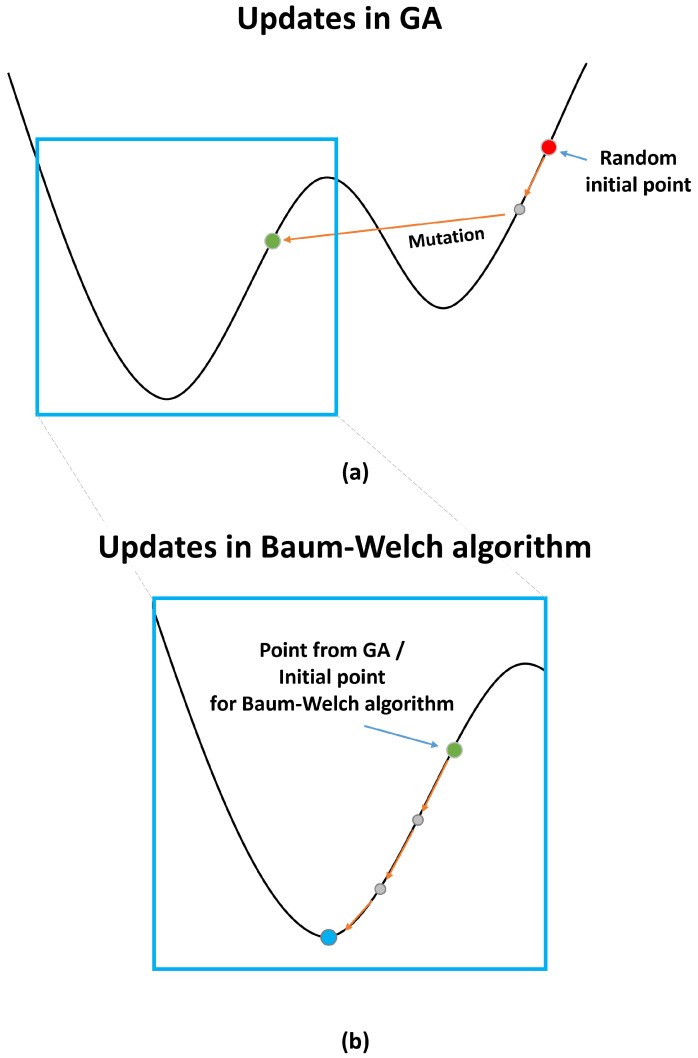
Parameters in HMM evaluated via sequential usage of GA and Baum–Welch algorithm: (**a**) GA yielded unbiased initial point for Baum–Welch algorithm in subsequent stage; (**b**) Baum–Welch algorithm yielded global optimal point from unbiased initial point.

**Figure 4 sensors-22-05088-f004:**
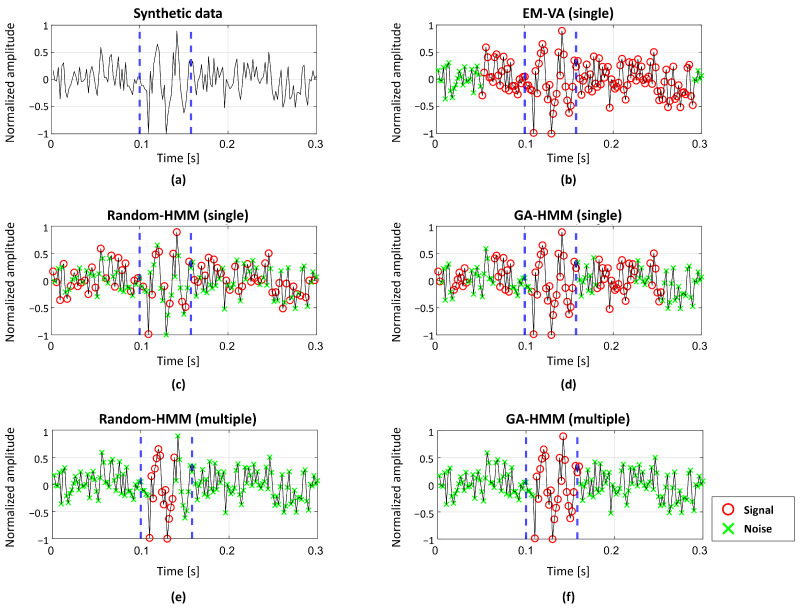
(**a**) Representative example of synthetic data with SNR of 8 dB. Detection results from (**b**) EM-VA, (**c**) single-measurement Random-HMM, (**d**) single-measurement GA-HMM, (**e**) multiple-measurement Random-HMM, and (**f**) multiple-measurement GA-HMM. The Interval for SOI is indicated by vertical dashed lines. Symbols “o” and “x” represent “signal” and “noise” states, respectively.

**Figure 5 sensors-22-05088-f005:**
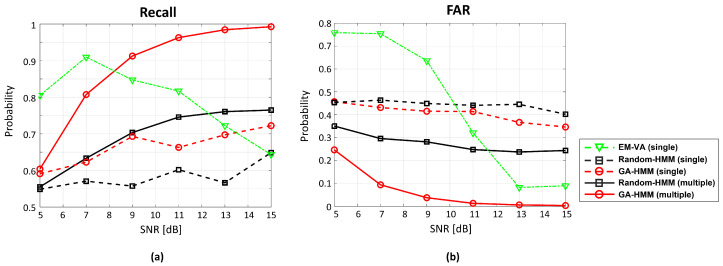
Detection performance of scheme according to SNRs: (**a**) Recall; (**b**) FAR. Multiple measurements consist of 30 synthetic data.

**Figure 6 sensors-22-05088-f006:**
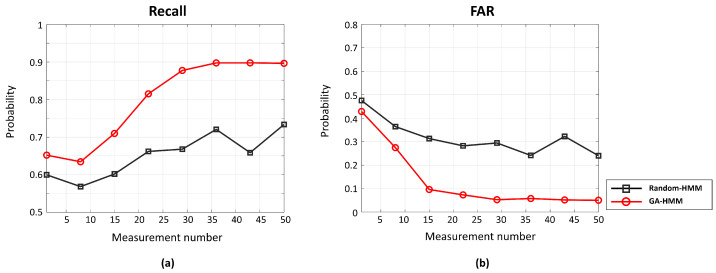
Detection Performance of scheme according to measurement number at fixed SNR of 8 dB: (**a**) Recall; (**b**) FAR.

**Figure 7 sensors-22-05088-f007:**
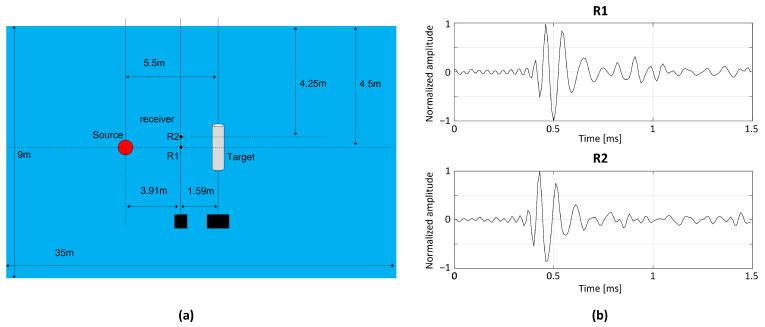
(**a**) Experimental environment; transmission signals are scattered by cylindrical shell and are received by two receivers at different water depths. (**b**) Portion of acoustic signals at two receivers after pulse compression (R1 and R2), which include intensive specular echoes and weak elastic surface waves.

**Figure 8 sensors-22-05088-f008:**
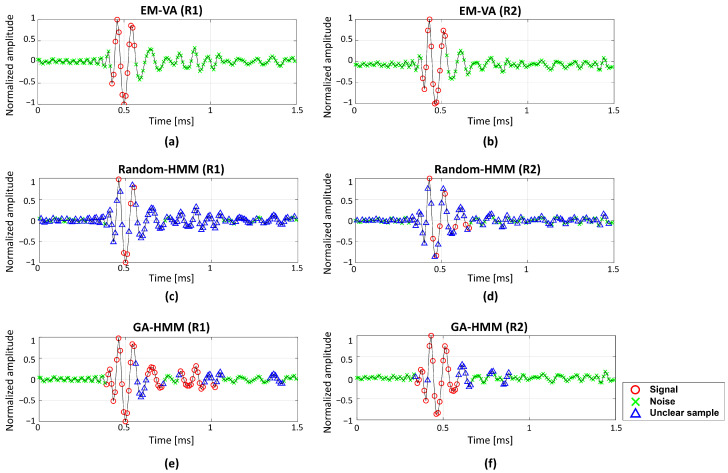
Detection results for signals measured by two receivers (R1 and R2). (**a**) Single-measurement detection result of EM-VA for R1; (**b**) single-measurement detection result of EM-VA for R2; (**c**) average detection result of multiple-measurement Random-HMM for R1; (**d**) average detection result of multiple-measurement Random-HMM for R2; (**e**) average detection result of multiple-measurement GA-HMM for R1; (**f**) average detection result of multiple-measurement GA-HMM for R2. Averaged values of the states from multiple-measurement Random-HMM and multiple-measurement GA-HMM are between 0 and 1. Samples exhibiting values exceeding 0.7 and less than 0.3 were classified as “signal” and “noise,” respectively. The remaining samples exhibiting values between 0.3 and 0.7 were “unclear samples”. Symbols of “o”, “x”, and “△” represent “signal”, “noise”, and “unclear samples,” respectively.

**Table 1 sensors-22-05088-t001:** Recall, false alarm rates, and computation time of the investigated schemes.

Scheme	Recall	False Alarm Rate	Computation Time
EM-VA (single)	0.86	0.63	0.83 s
Random-HMM (single)	0.60	0.48	0.08 s
GA-HMM (single)	0.65	0.43	6.76 s
Random-HMM (multiple)	0.69	0.27	11.02 s
GA-HMM (multiple)	0.88	0.06	12.10 s

## Data Availability

Not applicable.

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
