# Peer review of "Underwater Acoustic Signal Detection Using Calibrated Hidden Markov Model with Multiple Measurements"

_sensors, 2022, doi:10.3390/s22145088_

Round 1

Reviewer 1 Report

This paper proposed a signal-of-interest detection method using Hidden Markov Model(HMM). The originality of the paper is in two points: adoption of the genetic algorithm(GA) for parameter initialization of the Baum-Welch algorithm and modification of GA and HMM to apply multiple measurements. Overall, the paper is well-written and presented experimental results are clear to explain the superior performance of the proposed method. Nevertheless, some minor corrections and elucidation about some obscure parts are required.

1. What the abbreviation EV-VA stands for? The abbreviation was directly used without definition of the EV-VA.

2. In abstract and introduction, the proposed method is devied to reduces false alarm rates at a low signal-to-noise ratio and the presented experimental results were performed at over 5 dB SNR. Can you insist that over 5 dB SNR is low SNR condition in the aspects of signal detection?

3. By applying GA and multiple measurements to HMM, it is expected that the computational complexity of the entire system is increased. Can the proposed method perform in real-time? Please, compare the computational complexities of the proposed method and the competing methods.

4. Based on the Figure 4, it is seem that applying multiple measurements is further important than applying GA for peformance improvements. However, it is seem that in Figure 8 application of multiple mesurements is not so much effective as in Figure 4. Please answer the possible reason?

Author Response

The comments from the reviewer are responded in the attachment. Please, see the attachment. 

Reviewer 2 Report

In this paper, the Authors are proposing a method with favorable detection performance by using the hidden Markov model (HMM) for sequential acoustic data, which requires no separate training data.

The proposed method uses the genetic algorithm to reduce the sensitivity of the initial parameters

Some comparative tests for validation have been made. The detection results are compared with those from other detection schemes such as EM-VA and Random-HMM. They show that model achieves more accurate results.

After carefully reading, I find that this paper is extremely interesting, and the results very promising, however in order to further improve I would only recommend to improve the conclusions and more references on the background. (I suggest: doi: 10.3390/s22093415, doi: 10.3390/e23020135, doi: 10.3390/app11167530, doi: 10.3390/s21041425)

Author Response

(The authors gave the same response as above.)
